# Providing On-Site Laboratory and Biosafety Just-In-Time Training Inside a Box-Based Laboratory during the West Africa Ebola Outbreak: Supporting Better Preparedness for Future Health Emergencies

**DOI:** 10.3390/ijerph191811566

**Published:** 2022-09-14

**Authors:** Mostafa Bentahir, Mamadou Diouldé Barry, Kekoura Koulemou, Jean-Luc Gala

**Affiliations:** 1Centre for Applied Molecular Technologies (CTMA), Institute of Clinical and Experimental Research, Université Catholique de Louvain, Avenue Hippocrate 54-55, B1.54.01, B-1200 Brussels, Belgium; 2Laboratoire des Fièvres Hémorragiques Virales de Guinée, N’Zérékoré P.O. Box 50, Guinea; 3Laboratory of the Prefectural Hospital of Gueckedou, Gueckedou P.O. Box 82, Guinea

**Keywords:** ebola viral disease, just-in-time-training, BioFire FilmArray^®^ BioThreat-E, mobile laboratory tent, molecular diagnosis

## Abstract

The Biological Light Fieldable Laboratory for Emergencies (B-LiFE) is a box-based modular laboratory with the capacity to quickly deploy on-site in cases of uncontrolled spread of infectious disease. During the 2014–2015 West Africa Ebola outbreak, this tent laboratory provided diagnostic support to the N’Zerekore Ebola Treatment Center (ETC), Guinea, for three months. One of the objectives of B-LiFE deployment was to contribute, as much as possible, to national capacity building by training local scientists. Two Guinean biologists were selected according to their basic biological knowledge and laboratory skills among 50 candidate trainees, and were integrated into the team through “just-in-time training” (JiTT), which helped the biologists acquire knowledge and laboratory skills beyond their expertise. The JiTT program was conducted according to standard laboratory procedures, in line with international biosafety guidelines adapted to field conditions. Supervised acquisition of field-laboratory practices mainly focused on biochemical testing and Ebola viral load quantification using routine PCR-based detection, including the Biofire FilmArray^®^ system (BFA), a novel, as yet non-validated, automated assay for diagnostic testing of Ebola virus disease at the time of B-LiFE deployment. During the JiTT, the two biologists were closely involved in all laboratory activities, including BFA validation and biosafety procedures. Meanwhile, this successful JiTT enhanced the B-LiFE in-field operational capacity and contributed to national capacity building. A post-training evaluation and contacts were organised to assess the evolution and technical skills gained by the two researchers during the B-LiFE mission. At the end of the B-LiFE mission, both biologists were enrolled in follow-on programmes to curb the epidemic spreading in Africa. These results demonstrate that during infectious disease outbreaks or major crises, the JiTT approach can rapidly expand access to critical diagnostic testing and train local staff to do so.

## 1. Introduction

Epidemics are unpredictable and often deadly [1]. Despite worldwide improvements in public health over the past few decades, the emergence of new pathogenic agents with pandemic potential is expected to increase, favoured by factors such as global warming, increasing economic activity, global movement of people and goods, and high-density urban living. In addition, deforestation brings humans into contact with animals that act as new pathogenic vectors while armed conflicts cause the displacement of civilian populations, both of which create local health problems with a subsequent increase in the frequency and spread of infectious diseases [2,3].

National healthcare workers, including laboratory workers, need to be properly trained to deliver an effective response to emerging outbreaks; hence their early exposure makes them the first victims in the context of inadequate healthcare and laboratory infrastructure, including lack of training, information, and protective equipment [4,5]. In almost all past outbreaks of Ebola virus disease (EVD), the fatality rate among healthcare workers with documented infections was higher than that among non-healthcare workers [6]. Moreover, sudden-onset outbreaks undoubtedly increase the burden on countries already lacking diagnostic capacity [7]. Accordingly, “just-in-time training” (JiTT) is increasingly proposed as an efficient method to provide specific information, instructions, and guidelines where they are urgently needed, but often cannot be completed in due time. JiTT appears to be a non-expensive, efficient solution to rapidly provide healthcare and laboratory workers with new knowledge and skills during deadly outbreaks, as recently demonstrated by the response to the COVID-19 pandemic [8,9]

Pathogen identification is the fundamental component of a successful response to an epidemic, making diagnostic testing a national capacity building issue that urgently needs to be addressed. To this end, training in the use of new diagnostic tests is part of global JiTT for a timely emergency response. Providing diagnostic support and rapid training of biologists to correctly and safely perform these analyses is invaluable for moderately low- and middle-income countries [10]. Automated diagnostic assay platforms are commonly used for point-of-care (POC) or near-POC testing. Their on-site use and rapid turnaround times are considered a game-changer when an outbreak overcomes weakened or disrupted health services, and/or when demand for critical molecular diagnostics exceeds the existing national capacity [11,12,13,14]. The BioFire FilmArray^®^ BioThreat-E (BFA) test is one of several automated multiplex polymerase chain reaction (PCR) assays that have been proposed for syndrome-based, disease-specific diagnosis, and was proposed for EVD diagnostic testing during the 2014–2015 Ebola outbreak in West Africa [15,16,17]. However, at the time, BFA was novel, as yet non-validated, and its application in a laboratory tent was unprecedented.

The B-LiFE is a box-based modular laboratory tent that is rapidly deployable under field conditions in case of emergencies, such as an uncontrolled disease outbreak [18,19,20]. The use of B-LiFE can be activated through a bilateral agreement with a host nation, the European Civil Protection Mechanism (EUCPM), or the Global Outbreak Alert and Response Network—World Health Organisation (WHO-GOARN). The deployable laboratory has the capacity to provide molecular diagnostic services [21,22], mass screening [23], satellite communication, a local internet connection, as well as integrate data management and transfer through its laboratory information management system (LIMS) [24]. B-LiFE capabilities result from continuous development by the Centre for Applied Molecular Technologies of the Université catholique de Louvain.

This paper reviews the contribution of B-LiFE to JiTT during its operational deployment in Guinea and as EVD diagnostic support to the N’Zerekore Ebola Treatment Center (ETC) between December 2014 and March 2015, highlighting the potential role JiTT can play in national capacity building in response to infectious disease outbreaks.

## 2. Methods

### 2.1. Setting of the Mission and Mobile Laboratory Activation

The 2014–2016 Ebola outbreak in several West African countries, including Guinea, was of a very large scale and was unprecedented in terms of mortality and duration. Our mobile laboratory which was deployed in response to this health crisis was organised under the aegis and support of national and European institutions, including DG ECHO (European Commission DG for Civil Protection and Humanitarian Aid Operations) and ESA (European Space Agency). This action was also implemented in close cooperation with the WHO. However, this outbreak has evidenced how extremely difficult it was—and still is today—to activate the deployment of a mobile laboratory far abroad and to organise its logistics. The EU Civil Protection Mechanism (UCPM) was still in its early stages of development. Consequently, we needed the endorsement of several Belgian ministries for this deployment (the support of the Belgian armed forces for air transport; the agreement of the Ministry of the Interior, which oversees the activities of the Belgian civil protection, whose members supported our deployment on site, and the Ministry of Public Health). Obtaining permission from all these authorities, including answering the many questions about security guarantees at the proposed site of deployment of the ML, took several months of intense and lengthy discussions. Despite being ready, the B-LiFE team could not deploy until December 2014. The deployment was carried out with the logistic support of B-FAST (*Belgian First Aid and Support Team*). The mission came to an end when the epidemic in N’Zerekore, and in Guinea, ended in March 2015. The mission lasted until the end of the epidemic in N’Zerekore, and in Guinea, in March 2015.

### 2.2. Deployment Logistics

The B-LiFE was deployed in the forest area of Guinea Conakry, close to the city of N’Zerekore, Guinea, from December 2014 to March 2015 to support the local ETC (Figure 1). This healthcare facility was managed by Alliance for International Medical Action, a French non-governmental organisation. The laboratory team consisted of four laboratory experts and six non-laboratory members involved mainly in organisational aspects such as internet connectivity, communication, logistics, and decontamination (Figure 1a). The laboratory staff was rotated every four weeks, with staff overlapping for four days to ensure efficient handover of the laboratory and communication of all relevant operational information. 

### 2.3. JiTT Program

During our deployment in Guinea, a JiTT program was set up for local Guinean biologists who had volunteered to work with the B-LiFE staff. The first phase corresponded to a selection phase for the local candidates. Fifty volunteers applied for the JiTT program and declared themselves competent in biology. With the emergency and critical biosafety aspects in mind, it was essential to quickly and efficiently select those who were most qualified to perform laboratory work. The selection criteria were therefore based on both the results of a written test assessing their basic knowledge of molecular biology and biosafety fundamentals, as well as an interview to assess their professional background and practical laboratory skills, including possible participation in a previous similar mission. Among the 50 candidates, two local biologists were selected and joined the team. 

The JiTT began with a review of the standard operating procedures (SOPs) we had developed for our biosafety and biosecurity protocols and procedures as well as for the methodologies used in our mobile laboratory. The next step was a series of rehearsals of procedures and techniques by experienced laboratory staff. The last step was to directly involve them in carrying out the procedures and analyses while closely supervising them, followed by granting them complete autonomy while loosely supervising them. At that time, they were part of the laboratory staff and contributed to daily activities throughout the mission. The methodology followed to train the local biologists has been updated in the appropriate section.

### 2.4. Molecular and Biochemical Testing

The role of the B-LiFE was to perform rapid on-site screening for Ebola Zaire virus and malaria (*Plasmodium falciparum*) on blood samples from suspected Ebola cases and swabs taken from reported deaths in the community. We detected Ebola virus RNA and quantified the viral load using the RealStar Filovirus Screen RT-PCR Kit (Altona Diagnostics, Hamburg, Germany) with the CFX96 PCR instrument (Bio-Rad Laboratories, Hercules, CA, USA) according to the manufacturers’ instructions. We tested each sample in triplicate, with quantitative PCR (qPCR) results expressed as a quantitative cycle threshold (Cq). A Cq value flagged as “undetermined” by the thermocycler software after 40 cycles was defined as a negative result for EVD. We diagnosed malaria infection using a rapid lateral flow test (SD-Bioline, Standard Diagnostics Inc., Suwon-si, South Korea). We followed up with Ebola-positive patients to assess the efficacy of a 10-day oral treatment with Favipiravir^®^ (JIKI Trial; JIKI means “hope” in Kissi language, a Mel language of West Africa) [25]. This trial required reporting potential drug side effects, necessitating regular monitoring of vital biological parameters. Consequently, we used the handheld i-STAT blood analyser (Abbott Laboratories, Wavre, Belgium) to monitor the following parameters in whole blood with the CHEM8 + i-STAT cartridge (Abbott Laboratories): sodium and potassium electrolytes, ionised calcium, total CO_2_, haemoglobin, haematocrit, glucose, urea, and creatinine. We used the Piccolo Xpress automated clinical chemistry analyser (Abaxis Europe GmbH, Griesheim, Germany) as a back-up. The biosafety procedure for handling and using the iSTAT biochemical analyser was previously described [26]. We followed similar methods when loading 300 µL of patient blood sample into the Piccolo AmLyte 13 cassette (Abaxis, Union City, CA, USA) before loading it into the Piccolo Xpress device for measurement [27]. Biochemical analyses enabled the medical staff to correct fluid and electrolyte imbalances in Ebola patients. 

### 2.5. BFA Test Validation

During our deployment, we implemented the BioFire FilmArray^®^ BioThreat-E test (BioFire Diagnostics, Salt Lake City, UT, USA) for EVD diagnosis for the first time under field-laboratory conditions. As previously stated, the BFA had never been validated for EVD testing prior to the Ebola outbreak and our deployment in N’Zerekore. The B-LiFE team intended to use it for the first time as an indicator of a team’s ability to quickly learn a new emerging technology during an ongoing mission, and to teach trainees how to use it through a JiTT. The primary objective was to familiarise laboratory staff and trainees with the BFA test’s analytical steps. Prior to validating its use for EVD diagnosis, the preparatory work considered the constraints and requirements of the harsh field conditions and focused on the biosafety procedures associated with the BFA testing. We analysed a series of nasal swabs collected from B-LiFE volunteers using the BFA respiratory panel kit, which did not require any specific biosafety procedure. During this learning phase, we used a tent-environment exposed swab which serves as negative control. This negative control may not appear to be an optimal quality-control method, but it provided the B-LiFE staff and trainees with hands-on experience with the system and better prepared them for BFA testing of Ebola samples. The following step entailed validating the use of BFA for EVD diagnosis. There was enough material for BFA testing in 16 archived samples (11 plasma and 5 urine samples) that were stored at −20 °C. The obtained EVD diagnostic results were compared with previously acquired qPCR results for the same samples. Each sample was tested in triplicate. 

### 2.6. Biosafety Aspects

Members of the ETC medical team, wearing complete personal protection (PPE), obtained blood samples from every patient clinically suspected of EVD in accordance with WHO and CDC international guidelines. The guidelines differed for the laboratory staff who do not come into contact with patients but analyse their samples: in this case, the personal protective equipment included a single-use apron, a double pair of single-use gloves with long cuffs, and laboratory safety boots, without a mask. The difference in personal protective equipment between the ETC medical team and the B-LiFE staff is due to the nature of the biosafety procedures applied in the laboratory for the analysis of biological samples. These include decontaminating the surface of any sample to be analysed before bringing it into the laboratory, as well as inactivating any pathogens contained in that sample within a Class III biosafety cabinet (negative pressure; air filtration) before proceeding with the analysis of the sample outside the glovebox in the laboratory. These recommendations were followed by all mobile laboratories deployed during the Ebola crisis (Guinea, Sierra Leone, Liberia). 

We received all samples through a window in the internal laboratory fence (Figure 1b). For biosafety purposes, a specific, transportable, biosafety class III, negative-pressure glovebox (O.W.R. GmbH, Elztal, Germany; 0.75 × 0.50 × 0.45/0.35 m^3^) was used as a containment area to inactivate Ebola before any further sample processing (Figure 2a) [28]. Another identical glovebox was used to contain non-inactivated blood samples collected in lithium heparin-treated tubes, and to measure the blood parameters of Ebola patients with the i-STAT^®^ handheld device. Biochemical testing had to be carried out without the viral inactivation step, which is known to substantially affect blood analytes, thereby altering the biological results. To mitigate the risk of Ebola exposure due to contact with non-inactivated blood specimens in open tubes, strict biosafety procedures were put in place [26]. We proceeded with regular and thorough sodium hypochlorite decontamination of blood tubes, material, and equipment contained inside the glovebox according to the available guidelines [29]. During the last month of the mission, we also conducted biochemical monitoring using the Piccolo Xpress instrument due to unexpected technological shortcomings, which severely hampered the use of the i-STAT, as previously reported [26]. The larger size of the Piccolo Xpress prompted the installation of a larger plexiglass glovebox (Mbraun, Garching, Germany; size: 0.95 × 0.65 × 054/042 m^3^) (Figure 2b,c). The larger glovebox complied with the requirements for safe handling of the BFA cassettes and was therefore dedicated for validation of the BFA test for EVD diagnosis.

### 2.7. Waste Management

All delivery containers or tubes containing patient blood samples and buccal swabs that arrived at the laboratory, including those sent to look into community deaths, were decontaminated and handled inside a class III biosafety cabinet according to current procedures. Tips, tubes, and other single-use items that contained biological waste were decontaminated in the glovebox using a 1% chlorine solution before being disposed in a waste bin (Safesharp biohazard bin) [26]. At the end of the day or when the waste bin was almost full, it was hermetically sealed, the outside was decontaminated, and the bin was carefully taken out of the glovebox and placed in another larger biohazard container (yellow biohazard bins with an airtight closure). At the end of the day or when it was full, the container was finally sealed, decontaminated from the outside, and transported to the incinerator where the biowaste was incinerated by a specialised team in full Ebola protective clothing (Figure 3).

### 2.8. Ethical Approval

Ethical approval for the JIKI study was obtained and described previously [25]. Validation of the BFA test using human samples was approved by the Institutional Review Board of the Université catholique de Louvain/Saint-Luc Hospital (Brussels, Belgium).

## 3. Results and Discussion

We first briefed the two selected candidates about the specificities and constraints of the laboratory work, and then introduced them to good analytical and biosafety practices as part of their integration into the B-LiFE team. During the JiTT program, the trainees were given time to familiarise themselves with the biochemical and molecular analyses performed daily in the laboratory. The first step was to learn the procedures for safe sample reception (Figure 4a) and use of different gloveboxes (Figure 4b–e), followed by how to work with hazardous samples potentially contaminated with the Ebola virus at various workstations, including virus inactivation in the glovebox and viral RNA extraction and amplification, and finally, data validation and transfer of results. In parallel, both trainees learned to safely and reliably carry out molecular and biochemical tests. The JiTT program enabled the trainees to quickly acquire the necessary skills and become fully functioning members of the B-LiFE team, albeit with continuous but somewhat looser supervision. Within the framework of the JiTT program, and from the time they were selected, including the learning and training phase, the two local operators actively participated in the lab’s activities closely with the B-LiFE team members for a schedule of 7 days a week and 8 to 10 h per day for a total period of 75 days. Both trainees received certificates of attendance issued at the end of the JiTT program (Figure 4f). Both local biologists performed over 900 analyses, including qPCR (*n* = 459) on blood samples and swabs, *plasmodium falciparum* rapid tests (*n* = 112) as well as biochemical tests on non-inactivated lithium heparin blood samples using the i-STAT^®^ or Piccolo Xpress (*n* = 259) (Figure 2c and Figure 4e). A breakdown by EVD status of the total cases and samples analysed during the B-LiFE Ebola mission in Guinea is illustrated in Figure 5.

The Ebola outbreak in West Africa prompted the implementation of automated diagnostic assay platforms, including the BFA test, to address the critical need to reduce turnaround times and enable POC testing while maintaining high accuracy and reliability. During the JiTT program, the trainees were tasked with testing and evaluating the ease of use of the BFA test for EVD diagnosis in the field, and to validate its use in laboratory tent conditions. Considering that this was a previously unknown procedure for the laboratory staff, only a limited number of archived plasma and urine samples were tested. To prevent unexpected biosafety issues, the tests were performed towards the end of our mission when the diagnostic workload had significantly decreased. During this limited time, we selected 16 archived samples (11 plasma and 6 urine) which were available for the BFA test.

We devised a short preparatory study to familiarize both laboratory members and trainees with the analytical steps of the BFA test. During this learning phase, we used the BFA Respiratory Panel Kit (BioMérieux, Marcy-l’Étoile, France) to analyse a series of nasal swabs (*n* = 3) collected from B-LiFE volunteers and stored in universal transport media (UTM) (Copan, Brescia, Italy). In addition, we prepared a negative control swab by exposing the swab to the laboratory tent environment. This preparatory work allowed the laboratory team and trainees to become acquainted with BFA-specific biosafety procedures and analytical flow, including hydration of the cassette, preparation and loading of the sample in the glovebox, loading the cassette into the reader outside the glovebox, and use of the BFA software (Figure 6).

In conjunction with the B-LiFE team, the trainees implemented additional BFA-specific biosafety procedures for the analysis of Ebola samples. The method included the following steps: (i) thermal inactivation of viral samples according to the WHO-recommended protocol (60 °C for 60 min) using a thermoblock inside the glovebox, (ii) preparation of the cassette in the glovebox (i.e., hydration of the reagents, preparation, and loading of the sample), (iii) decontamination of the cassette inside the glovebox with 1% sodium hypochlorite, and (iv) transfer of the cassette outside the glovebox and loading it into the BFA analyser placed on a bench near the glovebox.

We compared the BFA results for sixteen archived samples (11 plasma and 5 urine samples) with previously acquired qPCR results for the same samples. Almost all of the BFA results (15/16) concurred with the qPCR results (Table 1). The only discordant result (i.e., BFA-negative/qPCR-positive) was observed in sample 04 collected from patient-1 in the convalescence phase. For this sample, 2 of 3 replicates were qPCR-positive, with a high mean Cq value (37.8), indicating a very low viral load. Although qPCR tests were only performed on fresh samples, all BFA tests were performed on archived samples stored at −20 °C with a mean interval period of 13.6 days (range: 3–46 days). Interestingly, performing BFA on previously frozen samples did not affect the quality of the results. Testing urine samples enabled us to evaluate the analytical efficiency of the BFA test on non-plasma body fluids, which is relevant since infectious Ebola virus is known to persist in semen, urine, and breast milk, with evidence of infectivity up to 531 days after initial diagnosis of infection [30,31]. Moreover, follow-up with patient-1 enabled us to compare BFA and qPCR test results on 7 successive plasma samples and 1 urine sample. Except for sample 04, as previously stated, the BFA and qPCR results agreed for all subsequent samples, demonstrating that the BFA test can be used for EVD diagnosis. Most importantly, the data confirmed that the BFA test was easy to implement and user-friendly for local trainees working in a laboratory tent. Therefore, this method can be used for rapid and reliable EVD screening under field conditions in remote areas with weakened, disrupted, or totally lacking laboratory resources and without highly trained personnel. Notwithstanding the reagent costs, this type of automated assay platform can quickly strengthen the diagnostic capacity of deployed laboratories, contributing to national capacity building in terms of identification and surveillance of endemic and epidemic agents [10].

Data on BFA testing for EVD diagnosis have become available [15,16,17] since our deployment in N’Zerekore. However, in contrast with our field conditions, previous validation studies were performed in fixed and reasonably well-equipped laboratory facilities: the first in the haemorrhagic fever laboratory of the Donka National Hospital (Conakry, Guinea) [15], the second in a small private hospital (Bo, Sierra Leone) [16], and the last at the Public Health England and the United Kingdom Ministry of Defence Laboratory at the Kerry Town ETC (Sierra Leone) [17]. Interestingly, while processing a much larger number of samples, Weller et al. used the same heat treatment step as in our laboratory tent to inactivate the virus [17]. Furthermore, all samples were tested by BFA within 6 days of the qPCR test, a significant difference from our time interval, which was up to 46 days [17]. However, a complete validation of the BFA test was not within the scope of the current study. Instead, we evaluated the user-friendly and biosafety features of fast, automated assays, using the BFA test as an example. The focus of the JiTT program was on hands-on testing and training of local non-expert collaborators with the aim of enhancing their expertise in laboratory practices and infectious disease diagnostics for future missions in their own country and other African countries, thus contributing, if only on a modest scale, to national capacity building in terms of individual laboratory competence.

Post-training evaluations and contacts were organised to assess the evolution and technical skills gained by the two researchers during the B-LiFE mission. To begin, we invited the two researchers to spend a week in our laboratory in Belgium a few months after the mission, depending on their availability. One of the researchers (KK) visited Belgium and worked in our laboratory for a week. We went over all of the molecular diagnosis procedures and techniques with him this week, allowing him to build on his knowledge from his participation in the JiTT in N’Zerekore.

We also conducted one-year phone interviews with both biologists to assess their career development and participation in other similar missions in their country, specifically, and in Africa in general. These interviews revealed that KK’s JiTT B-LiFE training enabled him to be hired by MSF Belgium/France. The second biologist (MDB) was appointed director of Guinea’s haemorrhagic fever laboratory, N’Zerekore. Rather than issuing a certificate of attendance in the future, it would be more useful to issue a certificate of technical competence, which could be more meaningful since this reflects the level of technical competence and the acquisition of essential quality assurance and biosafety principles.

## 4. Conclusions

JiTT enabled us to educate local biologists and implement practical, hands-on training during the B-LiFE Ebola mission in Guinea. By providing a range of new laboratory competencies beyond the expertise of local biologists, JiTT addressed the training needs relevant to local operational conditions while enhancing personal preparation and resiliency to future crises. Moreover, the JiTT program facilitated complete integration of the Guinean biologists within the laboratory team, maximising the overall operational capacity of the laboratory tent while minimising exposure to uncontrolled biological risks. It is noteworthy that both local trainees easily and rapidly learned to use a new automated platform for EVD diagnosis. Therefore, this JiTT approach and post-training evaluation may help expand access to critical diagnostic testing in the event of a major crisis. At the end of the mission, both local scientists engaged in follow-on programs to fight against the epidemic spread in Guinea as well as in African countries facing other types of outbreaks. These very positive results have helped us streamline the JiTT program and incorporate it into future B-LiFE on-site activities.

## Figures and Tables

**Figure 1 ijerph-19-11566-f001:**
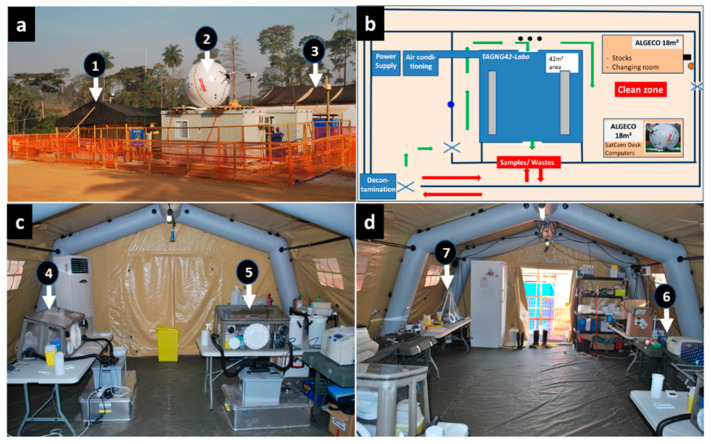
Organisation of the B-LiFE base of operation during the Ebola mission in Guinea. There were several modules consisting of (**a**) fully equipped inflatable laboratory tent (**1**), two containers for laboratory supplies and the command centre with a satellite internet network (**2**), and the Ebola treatment centre located inside the red zone and apart from the B-LiFE (**3**). (**b**) The laboratory tent, supported by a cement platform, is surrounded by two fences, one of which protects the overall perimeter of the laboratory and its auxiliary structures (including the power supply and air conditioning), and the other encloses the sensitive operational area (i.e., the laboratory tent and two Algeco containers). The tent and the two containers delineate the clean zone. The red arrows indicate the path followed by ECT personnel when bringing in samples, starting at a decontamination station (foot and hand washing) and ending at a “Samples/Wastes” window in the inner fence, opposite the front of the laboratory. Samples are passed to the laboratory staff through this window. The laboratory staff follows the same path to dispose of the waste (the laboratory wastes are systematically stored next to the window). The entrance and exit to the laboratory are located at the front of the tent. The green arrows show the path that the B-LiFE members must take to enter the tent or reach the clean zone. The blue and brown circles within the inner fence’s right and left borders indicate the location of the poles that support the outdoor Wi-Fi access points (router). The three black circles on the front of the tent represent water taps (chlorinated water). The uninterrupted power supply (UPS) is represented by the black rectangle to the right of the Algeco container in the top left corner. (**c**) Inside the laboratory tent, different gloveboxes were used for virus inactivation (**4**), biochemical testing (**c**) (**5**)—note that the large plexiglass glovebox was not in place at the time of the photo. (**d**) The extraction (**6**) and PCR (**7**) workstations were located at the entrance of the laboratory.

**Figure 2 ijerph-19-11566-f002:**
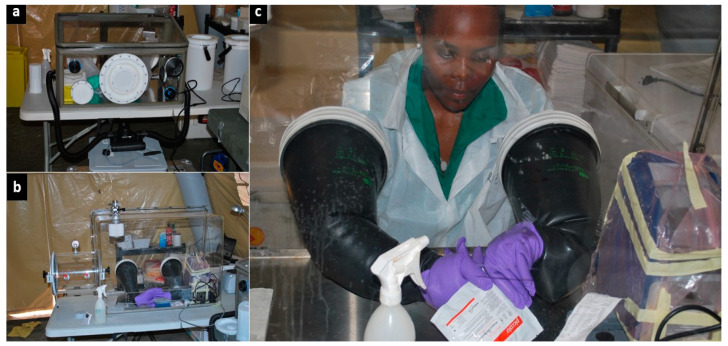
Two portable negative-pressure gloveboxes. (**a**) The smaller glovebox was used for inactive Ebola virus in patient samples, perform malaria tests, and monitor biochemical parameters with the handheld blood analyser. (**b**,**c**) The larger glovebox was used to monitor biochemical parameters with the Piccolo Xpress and to prepare the Biofire Filmarray cassette for Ebola diagnostics (not shown).

**Figure 3 ijerph-19-11566-f003:**
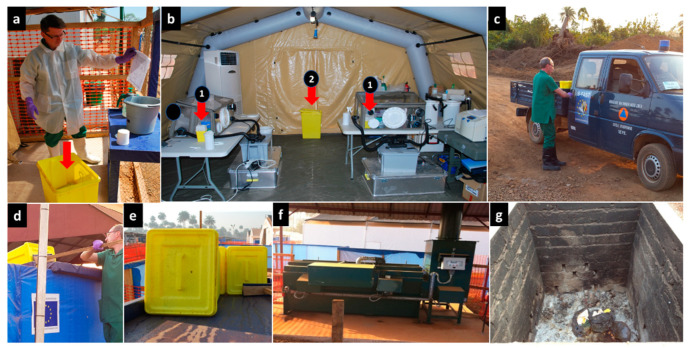
Illustration of the waste disposal route. It consisted of (**a**) waste management at the sample reception desk (Samples/Wastes window) outside the tent, (**b**) waste collection inside the laboratory tent, either in table top bin (**1**) or larger waste container (**2**) based on the activity and volume of the waste, (**c**) evacuation of waste by road to the incinerator, (**d**,**e**) delivery of waste containers over the ETC fence using a chute leading directly to the incinerator area. Depending on the nature of the wastes, a dedicated team wearing full PPE used an incinerator (**f**) or (**g**) a purpose-built pit.

**Figure 4 ijerph-19-11566-f004:**
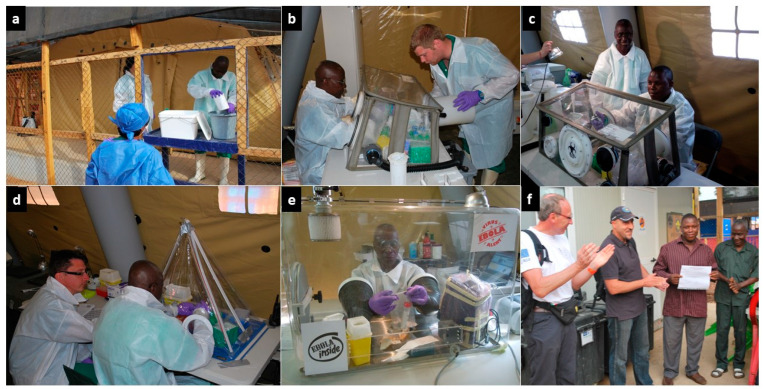
Biosafety training and qPCR procedures. A double fence surrounded the laboratory, the first to control entry to the site and perform biosafety measures (temperature monitoring, hand and shoe decontamination), the second to restrict access to the laboratory staff. (**a**) Samples were received through a window in the internal fence. The training focused on handling Ebola samples inside a negative-pressure glovebox, (**b**) first under supervision and then (**c**) autonomously, (**d**) to prepare a 96-multiwell plate for the qPCR inside a disposable glovebox, and (**e**) to perform biochemical tests. (**f**) Certificates of attendance were issued at the end of the JiTT program.

**Figure 5 ijerph-19-11566-f005:**
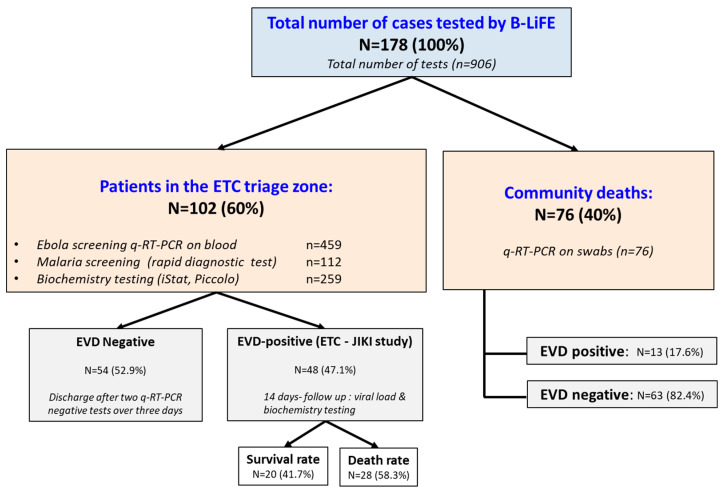
A breakdown by EVD status of the total cases and samples analysed during the B-LiFE Ebola mission in Guinea.

**Figure 6 ijerph-19-11566-f006:**
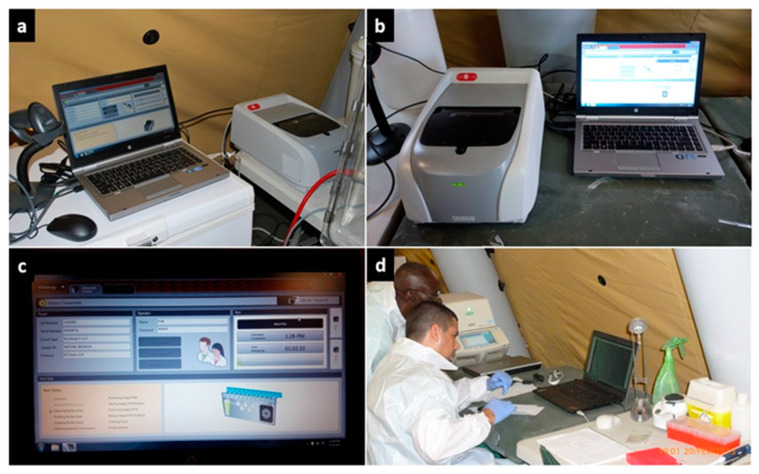
The BioFire FilmArray (BFA) system. (**a**) The BFA reader was installed inside the laboratory tent, (**b**) but outside the glovebox. (**c**) The BFA instrument was installed and connected to a computer with the adequate software-hardware. (**d**) The results generated by the trainees were transferred on a stand-alone computer to be reviewed and discussed with the staff.

**Table 1 ijerph-19-11566-t001:** Comparison of the BFA ^a^ and qPCR results ^b^.

Ebola Patient	BFA			qPCR Quantification Cycle (Cq)			Time Interval (Days)
	Sample	Test Date (Day-Month)	Result	Test Date (Day-Month)	Result	Mean Cq (SD)	
1	01	12-03	+	24-02	+	20.99 (0.03)	16
	02	11-03	+	26-02	+	26.15 (0.03)	13
	03	12-03	+	02-03	+	30.48 (0.13)	16
	04	12-03	-	05-03	+ ^c^	37.82 (3.51)	7
	05	12-03	-	06-03	-	-	6
	06	12-03	-	08-03	-	-	4
	07	13-03	-	09-03	-	-	3
	**08**	10-03	+	08-03	+	26.54 (0.04)	4
2	09	12-03	+	31-01	+	25.16 (0.07)	40
	**10**	11-03	-	05-03	-	-	6
3	11	11-03	-	23-02	-	-	17
4	12	12-03	-	23-02	-	-	18
5	13	11-03	+	24-01	+	27.52 (0.07)	46
6	**14**	13-03	+	26-02	+	26.28 (0.04)	15
7	**15**	13-03	+	26-02	+	27.14 (0.35)	15
8	**16**	14-03	-	09-03	-	-	5

^a^ BFA: BioFire FilmArray^®^ BioThreat-E test; ^b^ All results were generated in 2015. BFA tests were carried out on archived plasma and urine, while qPCR was performed on fresh whole blood and urine samples. Urine samples (08, 10, 14–16) are marked in bold. Ebola patient-1 recovered after 12 days of follow-up; ^c^ For sample 04, one of the three replicates was qPCR-negative. Samples were tested in triplicate.

## Data Availability

The datasets supporting the findings of this study are included in the paper.

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
