# Peer review of "Providing On-Site Laboratory and Biosafety Just-In-Time Training Inside a Box-Based Laboratory during the West Africa Ebola Outbreak: Supporting Better Preparedness for Future Health Emergencies"

_ijerph, 2022, doi:10.3390/ijerph191811566_

Round 1
Reviewer 1 Report
The aim of the paper is to present the benefits of a modular testing laboratory in underserved areas of Guinea experiencing the deadly Ebola outbreak in 2014-2015. The tent laboratory was utilized in a field area in conjunction with rapid point-of-care testing and just-in-time training (JITT) for proper specimen handling and testing for a limited time of 3 months. There is a need for publishing articles about rapid deployment of modular laboratories and the issues of staffing them with trained scientists in a sustainable way to promote national capacity. This is a focused study that is relevant to international public health and emergency preparedness.
The introduction nicely describes the necessity for these field laboratories, automated testing, and more efficient training of staff.
I feel that the article could benefit from more supportive description and data. JITT provides quick and effective training, yet quantitating the impact of JITT is understandably very challenging. To improve the paper, the following points should be addressed.
Why was this study limited to just 3 months of use of the modular laboratory, especially given the extent of the Ebola outbreak in Guinea?
While the photos of the modular laboratory are good, a diagram of the laboratory space could be helpful. A good reference article is: https://www.ncbi.nlm.nih.gov/pmc/articles/PMC5755746/pdf/pntd.0006135.pdf
What were the costs of setting up the modular laboratory with instrumentation, supplies, power sources and data management?
For Method section 2.2 JiTT Program, was the training all in-person or perhaps augmented with online training, or diagrams or hardcopies of procedures? Also, were the two biologists selected solely on their test scores or were there other criteria?
The article describes PPE, and the photos show the staff wearing a lab coat and gloves while there is no wearing of masks or face shields in the tent laboratory. What were the challenges of wearing facial protection?
Please briefly address waste management.
A diagram of the testing algorithm would be helpful. Another reference is: https://pubmed.ncbi.nlm.nih.gov/34860831/
In the Results and Discussion, what was the time, number of hours or days, necessary for the JiTT?
(Note for future events: would it be beneficial to perhaps issue certificates of technical competency, which could carry more significance and teach another level of quality assurance, instead of issuing certificates of attendance?)
Was there any post-training evaluation?
Would a cheek swab containing cellular material have been a better negative control than exposing the swab to test air?
The article nicely describes the details of testing with the BioFire FilmArray compared to the RT-PCR method.
Were there any analyses of the stability of the BFA testing cartridges under those field conditions?
The article contains data in Table 1 that is insufficient for a compete validation, and the authors justify the limited samples available during the time of this study. The number of BFA tests on patient samples beyond this validation study and perhaps used in this modular laboratory is unclear. What were the total number of tests on the BFA versus other platforms during this 3-month study?
Like the Abstract and Introduction, the Results and Discussion and Conclusions are well written, and make good, interesting points of value to the field. There are numerous self-citations in the References, although they are pertinent to the work.
Specific edits are as follows:
Lines 38-39: “emergence of new pathogenic and pandemic agents”
should be changed to “the emergence of new pathogenic agents with pandemic potential”
Lines 55: “where they are urgently needed, but cannot be delivered in due time.”
can be changed to “when they are urgently needed, but often cannot be completed in due time.”
Line 61-62: “part of global JiTT.”
should be changed to “part of global JiTT for a timely emergency response.”
Lines 97-98 Figure 1. Photo
should be changed to include the # (7) as indicated in the Figure legend Line 103 (7) workstations were located at the entrance of the laboratory.
Lines 102: “(4), biochemical testing (5) – note”
should be changed to “(4), biochemical testing (c) (5) – note”
Line 149: “was used to inactivate Ebola before any further sample processing”
should be changed to “was used as a containment area to inactivate Ebola before any further sample processing”
Lines 145-146: “Members of the medical team obtained blood samples from every patient clinically suspected of EVD wearing complete personal protection.”
should be changed to “Members of the medical team, wearing complete personal protection, obtained blood samples from every patient clinically suspected of EVD.”
Lines 150-152: We measured the blood parameters of Ebola patients inside another identical glovebox using the i-STAT® handheld device and non-inactivated blood samples collected in lithium heparin-treated tubes.
should be changed to “Another identical glovebox was used to contain non-inactivated blood samples collected in lithium heparin-treated tubes, and to measure the blood parameters of Ebola patients with the i-STAT® handheld device.
Lines 229: “UTM” transport media
should be changed to “universal transport media (UTM)
Author Response
Reviewer-1
The aim of the paper is to present the benefits of a modular testing laboratory in underserved areas of Guinea experiencing the deadly Ebola outbreak in 2014-2015. The tent laboratory was utilized in a field area in conjunction with rapid point-of-care testing and just-in-time training (JITT) for proper specimen handling and testing for a limited time of 3 months. There is a need for publishing articles about rapid deployment of modular laboratories and the issues of staffing them with trained scientists in a sustainable way to promote national capacity. This is a focused study that is relevant to international public health and emergency preparedness.
The introduction nicely describes the necessity for these field laboratories, automated testing, and more efficient training of staff.
I feel that the article could benefit from more supportive description and data. JITT provides quick and effective training, yet quantitating the impact of JITT is understandably very challenging. To improve the paper, the following points should be addressed.
Why was this study limited to just 3 months of use of the modular laboratory, especially given the extent of the Ebola outbreak in Guinea?
Authors’ response: We appreciate the reviewer's preciseness, very interesting suggestions for improving the paper, and nice appreciation of the value of our work.
The last comment about the 3-month mission is an excellent point that needs a thorough explanatory comment. It is absolutely true that the 2014-2016 Ebola outbreak in several West African countries, including Guinea, was of a very large scale and unprecedented in terms of mortality and duration. Our mobile laboratory was deployed in response to this health crisis was organised under the aegis and support of national and European institutions, including DG ECHO (European Commission DG for Civil Protection and Humanitarian Aid Operations) and ESA (European Space Agency). This action was also implemented in close cooperation with the WHO.
However, this outbreak has evidenced how extremely difficult it was - and still is today- to activate the deployment of a mobile laboratory far abroad and to organise its logistics. The EU Civil Protection Mechanism (UCPM) was still in its early stages of development. Consequently, we needed the endorsement of several Belgian ministries for this deployment (the support of the Belgian armed forces for air transport; the agreement of the Ministry of the Interior, which oversees the activities of the Belgian civil protection, whose members supported our deployment on site; and the Ministry of Public Health). Obtaining permission from all these authorities, including answering the many questions about security guarantees at the proposed site of deployment of the ML, took several months of intense and very frustrating discussions. Despite being ready, we couldn’t deploy until December 2014. The mission came to an end when the epidemic in N’Zerekore (...and in Guinea in general) ended in March 2015.
As said It should be noted that the activation of the deployment of mobile labs is still a complex issue in 2022 currently discussed with WHO / GOARN.
With the exception of this last comment, we have inserted this additional information as a new point “2.1 Setting of the mission and activation of the Mobile laboratory text” (text added lines 89-108)
While the photos of the modular laboratory are good, a diagram of the laboratory space could be helpful. A good reference article is:
Authors’ response: We have modified Figure 1 and added an annex figure showing the layout and architecture of the laboratory deployed in N'Zerekore. This plan shows the total area and the different workstations of the B-LiFE laboratory (Fig 1b).
The suggested article has been cited in the appropriate section and included in the list of references (ref 15).
What were the costs of setting up the modular laboratory with instrumentation, supplies, power sources and data management?
Authors’ response: The total cost of the mission was approximately €1.33 million, including equipment, insurance, transport and deployment of the laboratory to the site by C130 military aircraft, salary, accommodation and food for the researchers who worked seven days a week with a rotation of three teams of four researchers and a laboratory manager, as well as support staff for security, telecommunications and decontamination tasks.
These costs were largely covered by grants from the European Space Agency for the development of the B-LiFE mobile laboratory, as well as in-kind contributions from the Belgian National Defence and the non-governmental organisation ALIMA, which was in charge of building and running the Ebola treatment centre in N'Zerekore.
Because this was a humanitarian intervention, we felt it was inappropriate to include this information in the manuscript, unless otherwise stated by the reviewer.
We concentrated on training local researchers to enable knowledge transfer and to facilitate the development of an adequate national response for future epidemics.
For Method section 2.2 JiTT Program, was the training all in-person or perhaps augmented with online training, or diagrams or hardcopies of procedures? Also, were the two biologists selected solely on their test scores or were there other criteria?
Authors’ response: These points are really interesting. Let's go through them in the order we experienced them. The second question relates to the initial phase, which was a selection phase for a large pool of local candidates. Bearing in mind the emergency and the crucial biosafety aspects, we wanted to quickly and efficiently select those who were most qualified to perform laboratory work. The selection criteria were therefore based on both the results of a written test testing theoretical biology knowledge, and their professional background and laboratory experience (including possible participation in a previous similar mission). The section of the article dealing with this particular point has therefore been modified.
Regarding the first question, on-site training of candidates began immediately after their selection. It began with a review of the SOPs we had developed for our biosafety and biosecurity protocols and procedures as well as for the methodologies used in our mobile laboratory. The next step was a series of rehearsals of procedures and techniques by experienced laboratory staff. The last step was to directly involve them in carrying out the procedures and analyses while closely supervising them, followed by granting them complete autonomy while loosely supervising them. At that time, they were part of the laboratory staff and contributed to daily activities throughout the mission.
The methodology followed to train the local biologists has been updated in the appropriate section (lines 146-163).
The article describes PPE, and the photos show the staff wearing a lab coat and gloves while there is no wearing of masks or face shields in the tent laboratory. What were the challenges of wearing facial protection?
Authors’ response: Indeed, we explained in the "Biosafety Aspects" section that members of the medical team who treated Ebola patients and collected samples before sending them in triple packaging to the B-LiFE laboratory for analysis had to wear full personal protective equipment (PPE) in accordance with WHO and CDC international guidelines. The guidelines differed for the laboratory staff who do not come into contact with patients but analyse their samples: in this case, the personal protective equipment included a single-use apron, a double pair of single-use gloves with long cuffs, and laboratory safety boots, without a mask (the reason for this difference is twofold: (1) biosafety procedures include (1) decontaminating the surface of any sample intended for analysis before bringing it into the laboratory and (2) inactivating its contents inside a class III biosafety cabinet (negative pressure; air filtration) before proceeding with the analysis of the sample outside the glovebox in the laboratory. These recommendations were followed by all mobile laboratories deployed during the Ebola crisis (Guinea, Sierra Leone, Liberia).
The manuscript was modified at lines 213-225.
Please briefly address waste management.
Authors’ response: All delivery containers or tubes containing patient blood samples and buccal swabs, that arrived at the laboratory, including those sent to look into community deaths, were decontaminated and handled inside a class III biosafety cabinet according to current procedures. Tips, tubes and other single-use items that contained biological waste were decontaminated in the glovebox using a 1% chlorine solution before being disposed in a waste bin (Safesharp biohazard bin) (as detailed and illustrated in Irenge et al. 2017 – cf ref cited hereafter- ). At the end of the day or when the waste bin is almost full, it is hermetically sealed, the outside is decontaminated, and the bin is carefully taken out of the glovebox and placed in another larger biohazard container (yellow biohazard bins with a airtight closure). At the end of the day or when it is full, the container is finally sealed, decontaminated from the outside and transported to the incinerator where the biowaste is incinerated by a specialised team in full Ebola protective clothing.
A new paragraph (2.6 Waste management) describing the waste management procedure and a figure showing the different steps of this procedure (waste disposal route) are provided and included in the dedicated sections (lines 265-277))
REF 27 of the revised version : Irenge L, Dindart JM, Gala, JL. (2017). Biochemical testing in a laboratory tent and semi-intensive care of Ebola patients on-site in a remote part of Guinea: a paradigm shift based on a bleach-sensitive point-of-care device. Clinical Chemistry Laboratory Medicine, 2017; DOI 10.1515/cclm-2016-0456.
A diagram of the testing algorithm would be helpful. Another reference is: https://pubmed.ncbi.nlm.nih.gov/34860831/
Authors’ response: During this three-month Ebola mission in Guinea, local researchers in close collaboration with B-LiFE team members analysed several hundred (906) of samples from 179 suspected Ebola cases. While samples from 76 cases of community deaths were analysed to determine if death was due to Ebola virus infection, blood samples from103 suspected cases were screened for the virus contamination with a parallel rapid test for Plasmodium falciparum infection. Of these 103 suspected patients, 48 were confirmed positive and were admitted to the ETC for 14 days of treatment and monitoring of viral load (qPCR) and biochemical parameters (iStat and piccolo) every two days.
A new figure (Figure 5 in the updated version) has been prepared to show the distribution of the total number of cases and samples tested during the B-LiFE mission to Guinea according to their EVD status. The data are commented in the legend of the figure.
A diagram (Figure 5) illustrating the test breakdown is included in the manuscript in the appropriate section. The suggested reference is included in updated list of reference (ref 7).
In the Results and Discussion, what was the time, number of hours or days, necessary for the JiTT?
Authors’ response: From the time they were selected (including the learning and training phase) throughout the three-month mission, the two local operators B-Life trained actively participated in the lab's activities (21 December 2014 - 21 March 2015). In the first week of the mission, the recruitment process started. The two biologists closely worked with the B-LiFE team members for a schedule of 7 days a week and 8 to 10 hours per day for 75 days in total.
The appropriate section of the text is revised to include this information (303-306).
(Note for future events: would it be beneficial to perhaps issue certificates of technical competency, which could carry more significance and teach another level of quality assurance, instead of issuing certificates of attendance?)
Authors’ response: We fully agree with the reviewer. It is undoubtedly necessary to establish technical competence certificates for B-LiFE missions. We were unable to do so in Guinea due to the urgency and complexity of this mission. However, I wrote letters of recommendation with a technical assessment for each of the young Italian trainees during our deployment to Piedmont, Italy, for the COVID-19 crisis in 2021. Gala) wrote several letters of recommendation including a technical assessment for each young Italian trainee who participated to JiTT, as a support for their professional career. We must maintain this logic and standardise the process as much as possible.
This interesting comment was included in the revised version of the manuscript at lines 421 – 425.
Was there any post-training evaluation?
Authors’ response: Yes, post-training evaluations and contacts were organised to assess the evolution and technical skills gained by the two researchers during the B-LiFE mission. To begin, we invited the two researchers to spend a week in our laboratory in Belgium a few months after the mission, depending on their availability. One of the researchers (KK) visited Belgium and worked in our laboratory for a week. We went over all of the molecular diagnosis procedures and techniques with him this week, allowing him to build on his knowledge from his participation in the JiTT in N'Zerekore.
We also conducted one-year phone interviews with both biologists to assess their career development and participation in other similar missions in their country, specifically, and in Africa in general. These interviews revealed that KK's JiTT B-LiFE training enabled him to be hired by MSF Belgium/France. MDB was appointed director of Guinea's haemorrhagic fever laboratory, N’Zerekore.
This paragraph was added to the body text (410-421). A short sentence related to the post-training evaluation was also added to the summary (line 33-34).
Would a cheek swab containing cellular material have been a better negative control than exposing the swab to test air?
Authors’ response: The primary goal of this preparatory work was to familiarise the laboratory members and trainees with the analytical steps of the BFA test (which had never been used in our lab and had not been validated for EVD testing at the time of the mission). We used the BFA respiratory panel kit to analyse a series of nasal swabs collected from B-LiFE volunteers that did not require any specific biosafety procedure. Using a swab exposed to the tent environment may not have been the best negative control during this learning phase, but it gave us hands-on experience of how the system works and better prepared us for BFA testing of Ebola positive and negative samples.
The BFA corresponding paragraph was modified to insert these comments (lines 192-207).
The article nicely describes the details of testing with the BioFire FilmArray compared to the RT-PCR method. Were there any analyses of the stability of the BFA testing cartridges under those field conditions?
Authors’ response: No, this was not done during the mission. The BFA was only used for the JiTT, and did not produce results for the ETC. The comparison of the results with our gold standard (q-RT-PCR) was excellent, so we did not pursue further investigation.
However, the stability of the reagents was - and always is - tested for each method used as a clinical biology analysis during a mission.
This comment was not inserted in the revised version of the manuscript.
The article contains data in Table 1 that is insufficient for a compete validation, and the authors justify the limited samples available during the time of this study. The number of BFA tests on patient samples beyond this validation study and perhaps used in this modular laboratory is unclear. What were the total number of tests on the BFA versus other platforms during this 3-month study?
Authors’ response: In fact, as reported in the manuscript, the purpose of the study was not to perform a full validation of the BFA-test, but rather to use this new technology (not yet validated for EVD diagnosis in a mobile lab) as an indicator of the flexibility of the laboratory team, i.e. its ability to learn new technologies quickly during an ongoing mission, and to teach trainees how to use while respecting the new biosafety measures associated with this new technology.
As already stated in the manuscript, BFA-testing could only be conducted during the final 2 to 3 weeks of the mission when the diagnostic workload had significantly decreased.
During this limited time, only 16 archived samples (11 plasma and 6 urine) were available in total.
Prior to BFA testing, however, local trainees conducted performed 459 q-RT-PCRs, 112 plasmodium falciparum rapid tests and 259 biochemistry tests (either iStat or Piccolo biochemical readers) in close collaboration with B-LiFE team members as already detailed supra (as suggested by the reviewer, an algorithm describing the number of cases and tests performed throughout the mission has been added to the revised version of the article (Figure 5).
The body text was adapted accordingly lines 192- 207 and 338-339.
Like the Abstract and Introduction, the Results and Discussion and Conclusions are well written, and make good, interesting points of value to the field. There are numerous self-citations in the References, although they are pertinent to the work.
Authors’ response: All proposed specific edits have been implemented
Lines 38-39: “emergence of new pathogenic and pandemic agents”
should be changed to “the emergence of new pathogenic agents with pandemic potential”
Lines 55: “where they are urgently needed, but cannot be delivered in due time.”
can be changed to “when they are urgently needed, but often cannot be completed in due time.”
Line 61-62: “part of global JiTT.”
should be changed to “part of global JiTT for a timely emergency response.”
Lines 97-98 Figure 1. Photo
should be changed to include the # (7) as indicated in the Figure legend Line 103 (7) workstations were located at the entrance of the laboratory.
Lines 102: “(4), biochemical testing (5) – note”
should be changed to “(4), biochemical testing (c) (5) – note”
Line 149: “was used to inactivate Ebola before any further sample processing”
should be changed to “was used as a containment area to inactivate Ebola before any further sample processing”
Lines 145-146: “Members of the medical team obtained blood samples from every patient clinically suspected of EVD wearing complete personal protection.”
should be changed to “Members of the medical team, wearing complete personal protection, obtained blood samples from every patient clinically suspected of EVD.”
Lines 150-152: We measured the blood parameters of Ebola patients inside another identical glovebox using the i-STAT® handheld device and non-inactivated blood samples collected in lithium heparin-treated tubes.
should be changed to “Another identical glovebox was used to contain non-inactivated blood samples collected in lithium heparin-treated tubes, and to measure the blood parameters of Ebola patients with the i-STAT® handheld device.
Lines 229: “UTM” transport media
should be changed to “universal transport media (UTM)
Submission Date
04 August 2022
Date of this review
22 Aug 2022 23:35:18

Reviewer 2 Report
Your paper gives a valuable insight into a deployment of a field laboratory during the West African outbreak of Ebola. It is well written and contains much of interest to the biosafety and field laboratory community.
Some commnets
l146. Please describe Personal protection used
l149 define inactivation method used
l185 please define what JIKI stands for
Results
The JiTT is not properly described. It tells use what training they received but not the method of training used This needs to be expanded was it SOP based, how was the training shown to be successful. This needs to be expanded
Author Response
Reviewer-2
Your paper gives a valuable insight into a deployment of a field laboratory during the West African outbreak of Ebola. It is well written and contains much of interest to the biosafety and field laboratory community.
Authors’ response: We appreciate the reviewer's kind words about the JiTT work done inside the mobile lab and its description in this manuscript.
Some comments
l146. Please describe Personal protection used
Authors’ response: This point was also raised by reviewer n°1. The answer is therefore the same.
We explained in the "Biosafety Aspects" section that members of the medical team who treated Ebola patients and collected samples before sending them in triple packaging to the B-LiFE laboratory for analysis had to wear full personal protective equipment (PPE) in accordance with WHO and CDC international guidelines. The guidelines differed for the laboratory staff who do not come into contact with patients but analyse their samples: in this case, the personal protective equipment included a single-use apron, a double pair of single-use gloves with long cuffs, and laboratory safety boots, without a mask (the reason for this difference is twofold: (1) biosafety procedures include (1) decontaminating the surface of any sample intended for analysis before bringing it into the laboratory and (2) inactivating its contents inside a class III biosafety cabinet (negative pressure; air filtration) before proceeding with the analysis of the sample outside the glovebox in the laboratory. These recommendations were followed by all mobile laboratories deployed during the Ebola crisis (Guinea, Sierra Leone, Liberia).
For clarity, the type of PPE worn by trainees and laboratory personnel is described in the appropriate section of the revised manuscript (line 213-225).
l149 define inactivation method used
Authors’ response: In the revised manuscript, we have selected article references detailing the method of sample inactivation. (22,23 and 27).
l185 please define what JIKI stands for
Authors’ response: JIKI is the codename for the clinical trial titled "Efficacy of favipiravir in reducing mortality among Ebola virus-infected individuals in Guinea" (cited as .... in the revised manuscript). JIKI means hope in Kissi a Mel language of West Africa
This definition has been inserted in the revised version of the manuscript (line 176)
Results
The JiTT is not properly described. It tells use what training they received but not the method of training used This needs to be expanded was it SOP based, how was the training shown to be successful. This needs to be expanded
Authors’ response: A similar comment was made by reviewer n°1. We have updated the manuscript accordingly as follows:
The first phase corresponded to a selection phase for a large pool of local candidates. Bearing in mind the emergency and the crucial biosafety aspects, we wanted to quickly and efficiently select those who were most qualified to perform laboratory work. The selection criteria were therefore based on both the results of a written test testing theoretical biology knowledge, and their professional background and laboratory experience (including possible participation in a previous similar mission). The section of the article dealing with this particular point has therefore been modified.
The on-site training of candidates began immediately after their selection. It began with a review of the SOPs we had developed for our biosafety and biosecurity protocols and procedures as well as for the methodologies used in our mobile laboratory. The next step was a series of rehearsals of procedures and techniques by experienced laboratory staff. The last step was to directly involve them in carrying out the procedures and analyses while closely supervising them, followed by granting them complete autonomy while loosely supervising them. At that time, they were part of the laboratory staff and contributed to daily activities throughout the mission. A post-training evaluations and contacts were organised to assess the evolution and technical skills gained by the two researchers during the B-LiFE mission.
The methodology followed to train the local biologists has been updated in the appropriate section (lines 146-162)